# A Longitudinal Observational Study of Medical Cannabis Use and Polypharmacy among Patients Presenting to Dispensaries in Pennsylvania

**DOI:** 10.3390/biomedicines11010158

**Published:** 2023-01-08

**Authors:** Emily R. Hajjar, Allison Herens, Erin L. Kelly, Kayla Madden, Jessica M. Lungen, Brooke K. Worster

**Affiliations:** 1Jefferson College of Pharmacy, Thomas Jefferson University, Philadelphia, PA 19144, USA; 2Department of Medical Oncology, Sidney Kimmel Cancer Center, Thomas Jefferson University, Philadelphia, PA 19144, USA; 3Department of Family and Community Medicine, Thomas Jefferson University, Philadelphia, PA 19144, USA

**Keywords:** medical cannabis, polypharmacy

## Abstract

Background: Cannabis use is increasing among adults to treat a variety of health conditions. Given the potential for interactions and adverse events, it is important to assess the use of medical cannabis along with other concomitant medications when assessing for polypharmacy. Methods: The objective of this observational, longitudinal study was to examine medical cannabis (MC) use along with concomitant medications over 12 months in patients with serious medical conditions enrolled in the Pennsylvania (PA) Department of Health’s (DOH) Medical Marijuana Program and to collect and catalog which forms of MC patients are taking along with their concomitant medications. Results: There were 213 participants who completed the baseline surveys in full, and 201, 187, and 175 who completed the 1, 6, and 12-month follow-up surveys. The mean age of the participants was 41.3 years, and 54.5% were female. The mean number of MC products taken at baseline was 3.41 and 3.47 at the 12-month survey. Participants took an average of 3.76 (SD 3.15) medications at baseline and 3.65 (SD 3.4) at 12 months. Most commonly used concomitant medications at baseline included vitamins (42.3%), antidepressants (29.1%), analgesics (22.1%), herbal products (19.7%), and anxiolytics (17.8%). Conclusion: Participants used multiple medical cannabis products to treat a number of medication conditions in conjunction with multiple medications.

## 1. Introduction

Polypharmacy is the use of multiple medications and can be defined in many ways, with the most common definition being the use of five or more medications [1,2,3]. While polypharmacy is often needed to treat multiple, concomitant medical conditions, it is associated with negative outcomes such as falls, frailty, malnutrition, hospitalization, cognitive impairments, physical impairment, and increased mortality [2,3]. Traditionally, most polypharmacy research has studied prescription, over-the-counter medications, and complementary and alternative medications that patients get from pharmacies and has not included the use of cannabis products. 

More states have approved cannabis in medical or recreational capacities, and the use of cannabis among adults is increasing [4,5,6]. Daily or almost daily use (defined as 300 or more days in a year) has increased to nearly 4% of all US adults reporting use this frequently [7]. However, most clinicians report having infrequent conversations about cannabis use with patients and even less rarely documenting it in the medical record [8]. Commonly reported indications for using medical cannabis include anxiety, pain, appetite stimulation, and insomnia [9]. While more patients are using cannabis, little is known about how often it is used concomitantly with other medications. Use of cannabis has been associated with the risk of adverse events, overdose, cannabis use disorder, and pharmacokinetic and pharmacodynamic drug–drug interactions. [10,11,12]. With the rise of medical cannabis, it is important to regularly ask patients in all healthcare settings about the use of MC, regardless of medicinal or recreational purposes, when evaluating the potential risks and benefits of MC use for that specific patient. Given the fact that medical cannabis is often used concomitantly to treat conditions that can also be treated with prescription medications, our goal was to evaluate cannabis use patterns over the course of 12 months for patients enrolled in a state medical cannabis program. 

## 2. Materials and Methods

The objective of this observational, longitudinal study was to examine medical cannabis (MC) use over 12 months on polypharmacy in patients with serious medical conditions as defined by the Pennsylvania (PA) Department of Health’s (DOH) Medical Marijuana Program, as well as collect and catalog which forms of MC patients are taking and what their concomitant medications are. This study was conducted in partnership with Ethos Cannabis, a state-approved dispensary that has locations across PA. Informed consent, the baseline survey, and the follow-up surveys were administered over the phone and recorded by study staff in Qualtrics. Follow-up surveys occurred after one-, six-, and twelve-month periods. 

### 2.1. Participants 

Participants were recruited from dispensaries located throughout PA, through newspaper articles, and in local presentations to community members. Dispensary staff alerted patients to the study by providing flyers with all delivered and in-store purchases, posting information on their website, and verbally asking patients about their interest in participation. If patients agreed to be contacted, dispensary staff securely submitted their contact information to study staff on a weekly basis, or patients could contact study staff directly by phone or email. 

Patients were included in the study based on the following criteria: (1) age of at least 18 years old; (2) certification to use MC through the Pennsylvania Medical Marijuana Program, diagnosed by a healthcare provider and self-described as suffering from refractory symptoms or impaired quality of life despite previous medical management; (3) enrollment in the PA DOH Medical Marijuana Program; and (4) reported purchases of cannabis only through state-regulated dispensaries. The rationale for restricting purchases from state-regulated dispensaries was to enable two key aims for the larger, longitudinal study aimed at characterizing information related to MC products available in the PA dispensaries (pills, oils, topical formulations, liquids, and dry leaf or plant formulations for vaporization) and comparing this information to self-reported use [13]. Synthetic and prescription products such as dronabinol (Marinol^®^), nabilone (Cesamet^®^), and cannabidiol (Epidiolex^®^) were not included in this study as they are only available by prescription and are not available in medical marijuana dispensaries in Pennsylvania. Additionally, the rationale for why an individual uses cannabis on any given day may vary between use for medical or recreational purposes. Therefore, it is not practical to require participants to use the products in a strictly ‘medical’ sense. Exclusion criteria included: (1) known history of ongoing, active substance use disorder (including alcohol) and (2) pregnancy or breastfeeding. Female subjects were asked to sign a waiver attesting to not being pregnant or lactating. 

### 2.2. Measures 

#### 2.2.1. Demographics 

Demographics were collected at baseline using four items that asked participants their current age, ethnicity (*Latino or Hispanic*), race (*White*, *Black/African American*, *American Indian, Alaska Native*, *Asian*, *Native Hawaiian or Pacific Islander*, *or Other*), and gender (*Female, Male, Transgender female, Transgender male, or Other/non-binary*). 

#### 2.2.2. Medical Conditions 

At baseline, study personnel obtained the certifying medical condition(s) that brought patients to the dispensary. Next, participants were asked to report any other current medical conditions and the number of years since they were first diagnosed with each condition. In addition, the certifying medical condition recorded in the Pennsylvania Medical Marijuana Program database was also obtained through record review.

#### 2.2.3. Medications 

At baseline, participants self-reported any medications, supplements, or vitamins that they were actively using. For each follow-up, participants were asked if they were still taking each medication listed in their prior survey and to list any new medications that they were taking. For previously reported medications, participants were asked if the medication had been discontinued and if there had been any dosing changes. For each medication, participants were asked to report the corresponding medical condition and if the medication was related to their certifying condition.

The medications were categorized and recoded by a pharmacist and a pharmacy student on the study team and were then reviewed by a physician and a psychologist for any further clarification, with any discrepancies resolved through consensus among the three team members. The total number of medications used was calculated for each time point. For all follow-up surveys, the number of medications that were pre-existing, new, or stopped were calculated. For pre-specified medication categories (antidepressants, anxiolytics, benzodiazepines, opioids, sedatives/hypnotics, and stimulants; pre-existing medications), dosage changes were calculated as well (unchanged, increased, decreased, or discontinued).

#### 2.2.4. Medical Cannabis Use and Products 

At baseline, participants self-reported their MC use patterns and detailed what products they were currently using, if any. For each product, participants were asked to report the manufacturer, brand, formulation (*vaporization cartridge, flower, capsules, tincture, topical, patch, extract, suppository, RSO/edible oil, other*), route of administration (*oral, topical, inhaled, rectal*), CBD and THC percentages and ratios, and type (*indica, sativa, hybrid*). Participants were also asked to report the dosage (*number of inhalations, drops*, etc.) and frequency of use of each product. Participants were asked to report their daily use of each product (1 = *1 time per day up* to 7 = *7 or more times per day*) and how many days per week they used each product (1 = *less than once a week up* to 8 = *seven days a week*). 

In each follow-up survey, participants were asked if they were still using any of the products listed in the previous survey and, if so, for updated information on dosage and frequency. If participants reported they were no longer using a specific product, the reason for discontinuation was collected (*did not work, prefer other meds, too expensive, side effects, dispensary too far, availability, other*). Participants were asked to report any new products and all corresponding information about them and the participants’ use of them (brand, strain, formulation, route, CBD/THC percentages and ratios, type, dosage, and frequency). A count of total products was calculated as well as counts of discontinued and new products. 

During follow-up surveys, participants were asked if they had been using MC since their last survey *(yes—in the last week, yes—but not in the last week, yes—but not in the last month, no*). If participants reported that they had not used MC in the last month or since the last survey, the reason for discontinuation was collected (*did not work, prefer other meds, too expensive, side effects, dispensary too far, availability, other*). If participants responded that they used MC in the last week, it was followed by asking about the frequency of use in the last week (*never, sometimes, regularly*). Next, participants were asked if they had sought guidance about their MC products from their certifying physician or dispensary staff since the last survey (*yes, no*). If participants responded *no*, they were asked if they would be willing to talk to their certifying physician or dispensary staff about trying a different MC product (*yes, maybe, no*).

#### 2.2.5. Side Effects 

In all follow-up surveys, participants self-reported any side effects they experienced from MC since the last survey and rated each side effect on an intensity scale (1 = *mild* to 3 = *severe*). In all subsequent surveys, participants were asked if previously reported side effects continued and, if yes, to rate their severity. The counts of total current and discontinued side effects were calculated, as well as the average side effect severity.

#### 2.2.6. Symptoms and Quality of Life

Information on quality of life and the impact of MC on symptoms was collected. This information is reported in other publications. 

## 3. Results

There were initially 215 participants enrolled who were certified to use MC in PA for a serious medical condition between May and October 2020. A total of 594 individuals contacted research staff regarding their interest in participating in the study, of these 594 potential participants, 213 enrolled (2 did not complete the baseline), for a 35% response rate. Of the 379 individuals who did not enroll, reasons included: not meeting criteria (n = 18), declining or withdrawing once the study procedures were reviewed (n = 27), or losing contact after an initial response/no response to our initial outreach (n = 334). An additional 2 participants were not able to complete the baseline survey after initiating the consent process with study staff, and their data were not included in the final sample of 213 participants.

Of the 213 participants who enrolled and completed the baseline surveys in full, 201 participants were retained at their one-month follow-up, 187 were retained at their six-month follow-up, and 175 were retained at their one-year follow-up. Of those who did not complete the study, 35 were lost to follow-up, and 5 declined to participate further. 

The mean age of the participants was 41.3 years, with 54.5% being female (Table 1). The mean number of self-reported cannabis products taken at baseline was 3.41 and 3.47 during the 12-month survey (Table 2). Additionally, 55.4% of participants reported using MC via 1 route with 44.6% using MC in 2 or more routes (Table 2). The most common self-reported routes of administration were inhalation (93% at baseline, 86% at 12 months) and oral (44% at baseline, 37.7% at 12 months) (Table 2). A vast majority of patients continued to use cannabis throughout the study period in some form or another, as only 5 patients (2.9%) stopped using cannabis altogether at the 12-month survey. When looking at specific cannabis products, usage varied across the time points, with 55–78% of participants using a product previously reported at an earlier time point. In addition, 43–64% of participants reported using a new medical cannabis product compared to the survey before, and 69–78% of participants reported discontinuing a product. 

Participants took an average of 3.76 (SD 3.15; range of 0–10) medications in addition to their medical cannabis at baseline and 3.65 (SD 3.4; range 0–15) at 12 months. Additionally, 35.2% of patients at baseline and 31.4% of patients at the 12-month survey were taking five or more medications. The most commonly used concomitant medications at baseline included vitamins (42.3%), antidepressants (29.1%), analgesics (22.1%), herbal products (19.7%), and anxiolytics (17.8%) (Table 3). Those medication classes stayed consistent in terms of prevalence at the 12-month mark as well. 

Medications used to treat the same common indications as medical cannabis were analyzed to see if there was a potential change in usage over time. Those taking antidepressants and anxiolytics remained relatively stable on their medications and doses. In those taking an antidepressant, 69% remained on their original antidepressant, and 66.7% remained on the same dose at the 12-month mark, with 10% of the participants discontinuing their antidepressant at 12 months (Table 4). In those taking an anxiolytic, 71.4% remained on their original medication at 12 months, with 64% taking the same dose. This trend stayed consistent with opioids and sedative-hypnotics as well (Table 4). Of those taking a sedative/hypnotic, 80% remained on the same medication throughout the 12 months, with 92.3% staying on the same dose. Of those taking an opioid, 76.9% remained on their original medication, with 81.8% remaining on the same dose. 9.1% of participants were able to decrease their opioids dose, and 7.7% were able to discontinue their opioids at 12 months (Table 4).

The most commonly reported side effects of MC usage included dry mouth, increased appetite, and drowsiness/fatigue (Table 5). Those remained relatively constant across all time points. The most commonly reported reason for discontinuation of an MC product include availability, the MC product not being effect, and the preference to use another medication (Table 6). 

## 4. Discussion

The purpose of this survey was to assess participants’ medical cannabis usage within the Pennsylvania Medical Marijuana Program. The data collected in this survey provide needed information about the demographics, numbers of cannabis products used over time, routes of administration, and concomitant medication use of patients enrolled in a state-run medical cannabis program over the course of the year. 

This study highlights the fact that patients take multiple medical cannabis products along with multiple other medications. While the average number of concomitant medications was low, it is important to recognize that patients reported using up to 15 additional medications along with cannabis, and approximately 30% were taking 5 or more medications. Some patients may choose to use cannabis as a sole agent or as an adjunctive treatment to other medications, as evidenced by the proportion of patients concomitantly using opioids, benzodiazepines, and antidepressants. The use of medical cannabis can impact the levels of other medications and may have synergistic effects with patients also taking benzodiazepines, sedative-hypnotics, and opioids [14]. 

Approximately 40% of study participants did report using vitamins, and nearly 20% reported using complementary and alternative medications along with their cannabis. This may be an indication that some people are seeking more natural ways to treat conditions in addition to or as a replacement for prescription medications [15,16]. 

Another important aspect of the data is that participants exhibited high rates of starting new cannabis products and discontinuing others at each of the follow-up survey time periods. This could be due to patients needing to experiment with various products to find the one that works best for them, as patients exhibit different sensitivities to cannabis depending on the dose, dosage form, and prior use history [8]. This could also be due to product availability, as cannabis products may or may not be available over time depending on demand or crop production [8]. 

This study has several limitations. First, this study did attempt to look at the specific information for each MC product that participants reported taking. Unfortunately, it was evident during data collection that patient recall of specific information such as THC/CBD percent or ratios was hard to gather for each individual product. This may be due to the fact that people used various products at different times of the day and had a hard time recalling each one individually and the fact that MC products are often interchanged due to varied product availability over time. Patients were able to express that they use certain forms a certain number of times per day, but they may have had multiple products within the same dosage form. For example, patients could verbalize that they used a tincture multiple times per day, but they may have used one type of tincture in the morning and another later in the day depending on therapeutic effects they were looking to achieve or what types of adverse effects they were trying to avoid. For future studies, the challenge of self-reported data will impact how questions are asked with regard to getting accurate information on the types and composition of MC products used. The lack of participant recall also highlights the challenges of using MC in a clinical sense, in that patients could say they were using MC but did not know much more about the individual products. This also underscores the fact that clinicians who are trying to guide the patient’s choice of product may prefer to reference documented objective data (dispensing data) as opposed to self-report if small details are needed to help inform therapy decisions.

Another limitation was that study participants were recruited from dispensaries, which may have led to a sample bias toward those that heavily rely on cannabis for symptom management. This may have also impacted the number of people that stopped using cannabis altogether due to lack of efficacy or intolerable side effects, as a majority of patients continued to use MC products throughout the study. Data from a meta-analysis on the efficacy of cannabis in the treatment of pain found that 10% of participants withdrew due to adverse effects [17]. Other meta-analyses on cannabis in multiple sclerosis and chemotherapy-associated nausea and vomiting also found that study withdrawal rates were higher in those taking cannabis as opposed to a placebo [18,19].

Furthermore, the sample was mostly comprised of individuals who identified as white. This may be due to known differences in cannabis use by race [20]. Future surveys should focus on recruiting more diverse participants from both cannabis dispensaries and non-dispensary locations to get a sample more indicative of general use. Lastly, this study was conducted during the COVID-19 pandemic, which may have impacted patients’ MC and medication use due to potential social, economic, behavioral, physical, and mental health changes incurred as a result of this world-wide event [21,22]. 

## 5. Conclusions

Participants used multiple medical cannabis products to treat a number of medical conditions in conjunction with multiple prescription, over-the-counter, complementary, and alternative products. There is a high rate of variability from time point to time point as to what products people are using, so it is best to continually reassess patient use of medical cannabis products. 

## Figures and Tables

**Table 1 biomedicines-11-00158-t001:** Participant Characteristics.

Variable		Total (N = 213)
Gender	Male	94 (44.1%)
	Female	116 (54.5%)
	Other/Non-binary	3 (1.4%)
Age	Mean (SD)	41.3 (13.3)
	Range	18–78
Race	White	149 (70%)
	Black/African-American	33 (15.5%)
	Asian	6 (2.8%)
	Other/Prefer not to answer	24 (9.4%)
Ethnicity	Hispanic or Latino	20 (9.4%)
	Non-Hispanic/Latino	183 (85.9%)
	Prefer not to answer	10 (4.7%)
Already using cannabis for certifying condition	Yes (n, %)	165 (77.5%)
Use of benzodiazepine, opioid, or sedative	Yes (n, %)	51 (23.9%)
Certifying conditions via PA state database ^1^	Chronic Pain/Neuropathy	105 (49.3%)
	Anxiety	78 (36.6%)
	PTSD	33 (15.5%)
	Opioid Use Disorder	10 (4.7%)
	GI disorder (IBD, IBS, Crohn’s)	6 (2.8%)
	Cancer	3 (1.4%)
	HIV	2 (0.9%)
	Parkinson’s disease	1 (0.5%)
	Epilepsy/Seizure disorder	2 (0.9%)
Self-reported reason for use ^1^	Chronic Pain/Neuropathy	108 (50.7%)
	Anxiety	95 (44.6%)
	PTSD	45 (21.1%)
	Opioid Use Disorder	11 (5.2%)
	GI disorder (IBD, IBS, Crohn’s)	8 (3.8%)
	Depression	7 (3.3%)
	Arthritis	6 (2.8%)
	Insomnia	5 (2.3%)
	Seizure disorder	4 (1.9%)
	Fibromyalgia	2 (0.9%)
	HIV	2 (0.9%)
	Migraines	1 (0.5%)
	Cancer	1 (0.5%)
	ADHD	1 (0.5%)
	Multiple Sclerosis	1 (0.5%)
	Neurocognitive	1 (0.5%)
Symptom count	Median (Range)	4 (1–14)
Number of medical conditions reported ^2^	Median (Range)	3 (1–12)

Note. ^1^ Participants could list more than one condition; ^2^ Medical conditions related and unrelated to cannabis certification.

**Table 2 biomedicines-11-00158-t002:** Medical Cannabis Usage.

	Baseline (N = 213)	1 Month (N = 201)	6 Months (N = 187)	12 Months (N = 175)
	N (%)	N (%)	N (%)	N (%)
**Number of Participants taking a MC product**				
Existing Product (reported at a previous survey)	213 (100%)	157 (78.1%)	103 (55.0%)	116 (66.3%)
New Products		87 (43.3%)	121 (64.7%)	90 (51.4%)
Discontinued Products		140 (69.7%)	146 (78.1%)	125 (71.4%)
**Number of Self-Reported MC Products**				
	**Mean (SD)**			
Existing		1.82 (1.53)	1.13 (1.36)	1.48 (1.63)
New Products		1.53 (1.44)	2.35 (1.62)	1.99 (1.94)
Discontinued Products		1.55 (1.40)	2.01 (1.63)	1.80 (1.53)
Total Current Products	3.41 (1.52)	3.35 (1.81)	3.48 (1.89)	3.47 (2.45)
**Self-Reported MC Formulations**				
Inhalation	198 (93%)	185 (92%)	161 (86.1%)	151 (86.3%)
Oral	94 (44.1%)	81 (40.3%)	75 (40.1%)	66 (37.7%)
Topical	35 (16.4%)	36 (17.9%)	24 (12.8%)	20 (11.4%)
Suppository	1 (0.5%)	3 (1.5%)	4 (2.1%)	4 (2.3%)
**Number of Self-Reported Routes of Administration**				
1 route	118 (55.4%)	114 (56.7%)	109 (58.3%)	101 (57.7%)
2 routes	75 (35.2%	64 (31.8%)	59 (31.6%)	57 (32.6%)
3 routes	20 (9.4%)	21 (10.4%)	11 (5.9%)	9 (5.1%)
4 routes	0	0	1 (0.5%)	0

**Table 3 biomedicines-11-00158-t003:** Most Commonly Used Medications.

Medication Class	Baseline (N = 213)	1 Month (N = 201)	6 Months (N = 187)	12 Months (N = 175)
Vitamins	90 (42.3%)	80 (39.8%)	68 (36.4%)	62 (35.4%)
Antidepressant	62 (29.1%)	58 (28.9%)	54 (28.9%)	45 (25.7%)
Analgesic	47 (22.1%)	43 (21.4%)	31 (16.6%)	31 (17.7%)
Herbal product	42 (19.7%)	38 (18.9%)	31 (16.6%)	27 (15.4%)
Anxiolytic	38 (17.8%)	31 (15.4%)	25 (13.4%)	22 (12.6%)
Antihypertensive	36 (16.9%)	34 (16.9%)	27 (14.4%)	25 (14.3%)
Antihistamine	35 (16.4%)	34 (16.9%)	27 (14.4%)	25 (14.3%)
Anticonvulsant	30 (14.1%)	29 (14.4%)	24 (12.8%)	23 (13.1%)
Asthma/COPD medication	23 (10.8%)	21 (10.4%)	21 (11.2%)	22 (12.6%)

**Table 4 biomedicines-11-00158-t004:** Concomitant Medication Dosage Changes.

	1 Month	6 Months	12 Months
**Antidepressant Dosing Changes**	N = 62	N = 58	N = 54
Increased dosage	4 (4.8%)	7 (12.1%)	2 (3.7%)
Same dosage	58 (93.5%)	41 (70.7%)	36 (66.7%)
Decreased dosage	2 (1%)	3 (5.2%)	4 (7.4%)
**Anxiolytic Dosage Changes**	N = 38	N = 31	N = 25
Increased dosage	0	0	0
Same dosage	29 (76.3%)	19 (61.3%)	16 (64.0%)
Decreased dosage	2 (5.3%)	2 (6.5%)	3 (12.0%)
**Sedative/Hypnotic Dosage Changes**	N = 13	N = 13	N = 13
Increased dosage	0	1 (7.7%)	0
Same dosage	13 (100%)	8 (61.5%)	12 (92.3%)
Decreased dosage	0	0	0
**Opioid Dosage Changes**	N = 14	N = 14	N = 11
Increased dosage	0	1 (7.1%)	0
Same dosage	11 (78.6%)	5 (35.7%)	9 (81.8%)
Decreased dosage	2 (14.3%)	2 (14.3%)	1 (9.1%)

**Table 5 biomedicines-11-00158-t005:** Most Common Self-Reported Side Effects.

Side Effect	1 Month N (%)	6 Months N (%)	12 Months N (%)
Dry mouth	24 (11.9%)	21 (11.2%)	18 (10.3%)
Increased appetite	21 (10.4%)	33 (17.6%)	20 (11.4%)
Drowsiness/fatigue	19 (9.5%)	27 (14.4%)	16 (9.1%)
Anxiety	8 (4%)	12 (6.4%)	12 (6.9%)
Impaired mentation	7 (3.5%)	15 (8%)	9 (5.1%)
Cough	4 (2%)	7 (3.7%)	10 (5.7%)
Headaches/migraines	4 (2%)	6 (3.2%)	5 (2.9%)
Nausea/vomiting	4 (2%)	2 (1.1%)	1 (0.6%)
Dry eyes	3 (1.5%)	3 (1.6%)	2 (1.1%)
Lung/breathing problems	3 (1.5%)	3 (1.6%)	4 (2.3%)

**Table 6 biomedicines-11-00158-t006:** Reasons for discontinuation of a MC product.

	1 Month N (%)	6 Months N (%)	12 Months N (%)
Availability	95 (47.3%)	113 (60.4%)	74 (42.4%)
Not effective	15 (7.5%)	36 (19.2%)	12 (6.9%)
Prefer other medications	8 (4.0%)	4 (2.1%)	20 (11.5%)
Cost	8 (4.0%)	5 (2.6%)	3 (1.7%)
Side effects	5 (2.5%)	1 (0.5%)	3 (1.7%)

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
