# Peer review of "A Longitudinal Observational Study of Medical Cannabis Use and Polypharmacy among Patients Presenting to Dispensaries in Pennsylvania"

_biomedicines, 2023, doi:10.3390/biomedicines11010158_

Round 1

Reviewer 1 Report

Given that there are primarily two major cannabinoid compounds THC and CBD this study needs to look at THC and CBD%s in detail as THC is responsible for the high while CBD has medicinal effects in pain control.

Did the individuals changing their MC prescriptions increase their THC% meaning it was more for the high rather than pain control?

Also were synthetic Cannabinoid sprays used by any of the individuals. That needs to be a separate category.

Also Medicinal Cannabis is not a first line treatment for Depression, Anxiety and Severe Non-malignant pain. Were the other options tried and exhausted?

Author Response

Reviewer 1:

  1. Given that there are primarily two major cannabinoid compounds THC and CBD this study needs to look at THC and CBD%s in detail as THC is responsible for the high while CBD has medicinal effects in pain control.

Response: This study did attempt to look at the specific information for each MC product that participants reported taking.  Unfortunately, it was evident during data collection that patient recall of specific information such as THC/CBD percent or ratios were hard to gather for each individual product.  This may be due to the fact that people used various products at different times of the day and had a hard time recalling each one individually.  Patients were able to express that they use certain forms a certain number of times per day, but they may have had multiple products within the same dosage form.  For example, patients could verbalize that they used a tincture multiple times per day, but they may have used one type of tincture in the morning and another later in the day depending on that they were feeling or what types of adverse effects they were trying to avoid.

To improve the paper, we have added this as a limitation to point out that we did attempt to collect this information. (page 9)

  1. Did the individuals changing their MC prescriptions increase their THC% meaning it was more for the high rather than pain control?

Response: Medical cannabis use in the state of PA is not prescribed, but rather patients are certified to use MC for 23 approved medical conditions.  At the dispensary, patients are free to choose whatever products they think will work best for them and this student relied on self-reported information on MC products.

As stated above, it was our intention to try to track patients use of specific products over time, but we found early on that getting people to recall their use of specific MC products was quite challenging.  Future papers are going to examine the use of MC products on symptom management.    

  1. Also were synthetic Cannabinoid sprays used by any of the individuals. That needs to be a separate category.

Response: No, synthetic cannabinoid products were not included as part the medical cannabis products.  Information on page six has been added to clarify this point.    (page 2)

  1. Also Medicinal Cannabis is not a first line treatment for Depression, Anxiety and Severe Non-malignant pain. Were the other options tried and exhausted?

Response: This was an observation study looking at patients who were recruited based on their use of MC for treating condition such as depression, anxiety and pain.  Some patients were on concomitant prescription medications to treat their condition and others were not.  Since this was just an observation study, there was no mandate as to how patients treated their conditions.   In the state of PA, both anxiety and pain are indications for certification for cannabis use. 

Reviewer 2 Report

Both in the title of the manuscript and in the abstract there is no information about the aim of the work. Is it the estimation of the level of polypharmacy, including the share of cannabinoids or interactions between individual substances and cannabis? The Authors should clearly state in which direction the research was conducted. The title is too general.

The Authors should use either reference numbers or names depending on what the journal requires. They cannot use double nomenclature, e.g. lines 27, 30, 35.

Lines 49-52 the goal is too general, it's hard to deduce what the Authors meant.

The purpose of the manuscript given in lines 54-57 does not quite agree with that in lines 207-208. In my opinion, the Authors have a problem with clearly defining what their goal was. It should be clearly defined and the manuscript and title should be reworded accordingly.

In my opinion, the article does not meet the aims and scopes of Biomedicines .

Author Response

Reviewer 2:

  1. Both in the title of the manuscript and in the abstract there is no information about the aim of the work. Is it the estimation of the level of polypharmacy, including the share of cannabinoids or interactions between individual substances and cannabis? The Authors should clearly state in which direction the research was conducted. The title is too general.

Response: The title and the abstract have been updated to provide a clearer understanding of the type and objectives of the study. (page 1)

  1. The Authors should use either reference numbers or names depending on what the journal requires. They cannot use double nomenclature, e.g. lines 27, 30, 35.

Response: Thank you for pointing this out, the authors names have been removed, and citation numbers are used for references.

  1. Lines 49-52 the goal is too general, it's hard to deduce what the Authors meant.

Response: We have modified the text with more specific information regarding the risks of cannabis use to convey our message of considering the use of cannabis along with other medications.

  1. The purpose of the manuscript given in lines 54-57 does not quite agree with that in lines 207-208. In my opinion, the Authors have a problem with clearly defining what their goal was. It should be clearly defined and the manuscript and title should be reworded accordingly.

Response: We agree that the intent was stated differently at different points in the manuscript and appreciate this being pointed out to us.  We have clarified all places to reflect that the main intent was the study cannabis use patterns, to collect and catalog what forms of MC patients were taking and what concomitant medications patients were taking. 

  1. In my opinion, the article does not meet the aims and scopes of Biomedicines .

Response: This manuscript was submitted after discussion with Dr. Raup-Konsavage, guest editor, for the special issue on the “Therapeutic Potential for Cannabis and Cannabinoids”.

Reviewer 3 Report

A descriptive study, without clear and explained results

It talks about interactions, but it is not checked if there are potential interactions

Polypathology justifies polytherapy (it's normal), but it's not polypharmacy according to the definition

Although the follow-up was done at 6 and 12 months, there are no complete data presented for 6 months

Nothing is said about doses

Where are the data on the number of medicines? It is said in the text that we have a small number of associated drugs

We are talking about the risk in elderly patients, but in the study they have an average of 41.3

It is not specified which supplements and OTC you are talking about

10 of the 17 bibliographic references are used in the introduction, the total number of references being small

Many parameters mentioned in the methodology are not found in the results

Why did patients stop using certain compounds?

What adverse reactions occurred with MC ?

From the presentation of the results, it is not understood that the administration of those drugs is concurrent

The discussions are superficial

Author Response

Reviewer 3:

  1. A descriptive study, without clear and explained results

Response: Thank you for letting us know that more detail is needed.  We have added more detail throughout the manuscript to provide a clearer indication of the purpose of the study.

  1. It talks about interactions, but it is not checked if there are potential interactions

Response: We have removed the main points that stressed drug interactions as the data collected on the type of cannabis products and how often people were using each individual product was harder to capture than we had organically anticipated.  We had wanted to be able to provide more data on drug interactions, but given the availability of self-reported product data, that was no longer able to be done.

  1. Polypathology justifies polytherapy (it's normal), but it's not polypharmacy according to the definition

Response: We agree that polypathology may create polytherapy.  Polypharmacy can be defined in many ways, regardless of the number of concomitant disease states, but the most common definition provided in the literature is 5 or more medications.

  1. Although the follow-up was done at 6 and 12 months, there are no complete data presented for 6 months

Response: Thank you for pointing this out, we have added 6 and 12-month data where applicable.

  1. Nothing is said about doses

Response: As stated above, it was very difficult to gather information on dosing due to the self-reported aspect of data collection.  We attempted to gather data of %/ratio of THC/CBD and dosing for each product but patients were not able to recall in that much detail.  Furthermore, it was very hard to discuss dosing for inhalation products where no metered dose preparations are available. 

  1. Where are the data on the number of medicines? It is said in the text that we have a small number of associated drugs

Response: Page 5 (lines 193-198) contains the data for the concomitant number of medications taken along with MC products.  Most commonly prescribed medications are also available in Table 5.

  1. We are talking about the risk in elderly patients, but in the study they have an average of 41.3

Response: We deleted the statement about older adults as some of the references discussed the risks of polypharmacy in general and not just the geriatric population.

  1. It is not specified which supplements and OTC you are talking about

Response: We are not sure what you are referencing here in terms of supplements and OTC products.

  1. 10 of the 17 bibliographic references are used in the introduction, the total number of references being small

Response: More references have been added throughout the manuscript. 

  1. Many parameters mentioned in the methodology are not found in the results

Response: We appreciate you pointing this out.  T0 improve the manuscript we have added in more data with regard to self-reported side effects and reasons for discontinuation of MC products.  (tables 4 and 5; page 8)

  1. Why did patients stop using certain compounds?

Response: Thank you for pointing this out, we have added self-reported data as to the reason why people discontinued using MC products. (table 5)

  1. What adverse reactions occurred with MC ?

Response: We have added a table to address this missing information.  (table 4, page 8)

  1. From the presentation of the results, it is not understood that the administration of those drugs is concurrent

Response: The use of concurrent is that the medications were on the participant’s medication list at the same time the participant was using MC.  Concurrent does not mean they were taken at the same time of day. 

  1. The discussions are superficial

Response: Thank you for pointing this out.  More information has been added to the discussion, particularly with the limitations section to provide more depth.  (page 9)

  1. Summary: The authors present data on polypharmacy among medical cannabis patients presenting to dispensaries in Pennsylvania (a cohort followed over 12 months). The topic is very timely and the data are interesting, but I think there are some clear areas for improvement in the results.

Response: We appreciate how some information came across as vague and we have added detail to improve the overall results of the manuscript.

Reviewer 4 Report

Summary: The authors present data on polypharmacy among medical cannabis patients presenting to dispensaries in Pennsylvania (a cohort followed over 12 months). The topic is very timely and the data are interesting, but I think there are some clear areas for improvement in the results.

Major issues:

1.     The results described in text do not seem to match the data presented in the tables, which makes reading this a bit confusing. For example, the result in lines 172-173 (“55.4% of participants reported using MC via 1 route 172 with 44.6% using MC in 2 or more routes”) does not seem to be in Table 2 (there’s no “number of self-reported MC formulations” row). This is even more so the case for Table 3 data – the table presents just frequency of each drug class at baseline and 12 months, but there’s lots of other results presented in the text.

2.     Lines 189-190 “Medications used to treat common indications as medical cannabis were analyzed to see a potential change in usage over time.” – there is no mention of any statistical analysis conducted here. What is the authors’ rationale for not conducting a formal statistical test here? Similar to my first point, I feel that the data presented in this paragraph should be captured in a table.

Minor issues:

3.     Title – I think the title could be more specific and informative (maybe something like “Medical Cannabis Use and Polypharmacy among Patients Presenting to Dispensaries in Pennsylvania”; could also mention it’s a longitudinal cohort in the title as well)

4.     Introduction – I think it’s important to discuss specifically the literature regarding CBD and drug-drug interactions, as this helps to contextualize why there might be drug-drug interactions with medical cannabis products (this is maybe less important here if the authors plan to write another manuscript that would focus more on the THC and CBD concentration of the products used)

5.     Methods, line 83 – remove the term “abuse” as this term is stigmatizing. Instead, the authors should state the definition of problematic substance use used to evaluate this criterion

6.     Methods, line 119 – the authors mention they collected THC and CBD % and ratios, but I don’t see this data presented; is this going to be presented in another manuscript or is there another reason for not presenting these data?

7.     Methods – I notice there’s actually a fair amount of data in the methods not presented (quantity and frequency of MC use; reasons for discontinuing products; side effects) – the authors should clarify why certain data are presented here and not other.

8.     Results, line 172 – it seems to me that the # cannabis products at 12 months reported in the text (2.68) doesn’t match table 2 (3.47)

Author Response

Reviewer 4:

Major issues:

  1. The results described in text do not seem to match the data presented in the tables, which makes reading this a bit confusing. For example, the result in lines 172-173 (“55.4% of participants reported using MC via 1 route 172 with 44.6% using MC in 2 or more routes”) does not seem to be in Table 2 (there’s no “number of self-reported MC formulations” row). This is even more so the case for Table 3 data – the table presents just frequency of each drug class at baseline and 12 months, but there’s lots of other results presented in the text.

Response: To reduce the complexity of the tables, some information was presented in tabular format and other information was presented in the text.  As requested by your comments, to increase the clarity of this information, we have added in more information to the tables. (page 7)

  1. Lines 189-190 “Medications used to treat common indications as medical cannabis were analyzed to see a potential change in usage over time.” – there is no mention of any statistical analysis conducted here. What is the authors’ rationale for not conducting a formal statistical test here? Similar to my first point, I feel that the data presented in this paragraph should be captured in a table.

Response: No statistical tests were used to analyze the change over time.  To help provide more clarity, we will also include this information in a tabular format. (new table 3)

Minor issues:

  1. Title – I think the title could be more specific and informative (maybe something like “Medical Cannabis Use and Polypharmacy among Patients Presenting to Dispensaries in Pennsylvania”; could also mention it’s a longitudinal cohort in the title as well)

Response: Thank you for pointing this out.  We have updated the title to reflect more detailed information.  (page 1)

  1. Introduction – I think it’s important to discuss specifically the literature regarding CBD and drug-drug interactions, as this helps to contextualize why there might be drug-drug interactions with medical cannabis products (this is maybe less important here if the authors plan to write another manuscript that would focus more on the THC and CBD concentration of the products used)

Response: Due to limitations in data collection, the assessment of drug interactions was not able to be fully researched.  Therefore, we have decided to remove the emphasis on drug interactions as the results do not support the information stressed in the first draft of the manuscript.

  1. Methods, line 83 – remove the term “abuse” as this term is stigmatizing. Instead, the authors should state the definition of problematic substance use used to evaluate this criterion

Response: We appreciate you pointing this out.  The wording has now been changed to substance use disorder. (page 3)

  1. Methods, line 119 – the authors mention they collected THC and CBD % and ratios, but I don’t see this data presented; is this going to be presented in another manuscript or is there another reason for not presenting these data?

Response: As stated above, it was our intention to collect this data, but use of self-report was very limiting in terms of collecting this level of detail.  We have now acknowledged how hard this was in the limitations section of the discussion.

  1. Methods – I notice there’s actually a fair amount of data in the methods not presented (quantity and frequency of MC use; reasons for discontinuing products; side effects) – the authors should clarify why certain data are presented here and not other.

Response: Thank you for pointing this out.  Given the difficulty in collecting the quantity and frequency of use, this data cannot be presented. However, we have added data on the % of side effects as well as the reasons for MC product discontinuation. (page 8)

  1. Results, line 172 – it seems to me that the # cannabis products at 12 months reported in the text (2.68) doesn’t match table 2 (3.47)

Response: Thank you for pointing this out.  There was an error in the data reporting which has now been corrected.

Round 2

Reviewer 2 Report

In my opinion the manuscript has been sufficiently improved to be publicated in Biomedicines

Author Response

Thank you for taking the time to review the manuscript again.  We appreciate your comments in helping us improve this paper. 

Reviewer 3 Report

R 91 dronabinol instead of dronabinal

R165-169 the above information is repeated

Table 3 Opioid Dosage Changes, Decreased dosage , 1 mounth. Please check the pourcentage (is not 13,3% is 14,3%)

Table 5 Medication Class. The ATC classification would have been more appropriate because confusions can occur (anticonvulsants used as analgesics, antidepressants used as anxiolytics)

Author Response

R 91 dronabinol instead of dronabinal: Thank you for pointing this out, this typo has now been corrected. 

R165-169 the above information is repeated; We cannot find where this information is repeated in the manuscript. 

Table 3 Opioid Dosage Changes, Decreased dosage , 1 mounth. Please check the pourcentage (is not 13,3% is 14,3%); Thank you for pointing this out, we have corrected this mistake. 

Table 5 Medication Class. The ATC classification would have been more appropriate because confusions can occur (anticonvulsants used as analgesics, antidepressants used as anxiolytics); Thank you for pointing this out, we will take this into consideration for future studies. 

Reviewer 4 Report

I appreciate the time the authors took to respond to my comments and I feel all of my concerns have been addressed. I have no further comment.

Author Response

(The authors gave the same response as above.)
